# Chest-OMDL: Organ-specific Multidisease Detection and Localization in Chest Computed Tomography using Weakly Supervised Deep Learning from Free-text Radiology Report

**Xuguang Bai**[*1]                                   MEIXIXIXI217@GMAIL.COM
**Mingxuan Liu**[*1]                                   ARKTISX@FOXMAIL.COM
**Yifei Chen**[1]                                      JUSTLFC03@GMAIL.COM
**Hongjia Yang**[1]                          YANGHJ23@MAILS.TSINGHUA.EDU.CN
**Qiyuan Tian**[†1]                              QIYUANTIAN@TSINGHUA.EDU.CN
[1] *School of Biomedical Engineering, Tsinghua University, Beijing, China*

**Editors:** Accepted for publication at MIDL 2025

## Abstract

Deep learning (DL) models designed to detect abnormalities in chest computed tomography (CT) reduce radiologists' workload. However, training multidisease diagnostic models requires large expert-annotated datasets, significantly increasing model development cost. To address this challenge, we propose a weakly supervised learning (WSL) framework entitled Chest-OMDL for Organ-specific Multidisease Detection and Localization in chest CT. Chest-OMDL trains DL models using disease labels extracted by RadBERT from free-text radiology reports and multi-organ segmentation masks generated by the Segment Anything by Text (SAT) model, therefore reducing the need for manual annotation. Specifically, Chest-OMDL employs a Y-shaped Mamba model (Y-Mamba), comprising a feature extractor, an organ segmentation decoder, and a disease anomaly map generator. By incorporating multidisease anatomical knowledge, Y-Mamba is trained with a multi-task loss for organ-level weak supervision. Chest-OMDL was trained and validated on the large-scale CT-RATE dataset (25,692 non-contrast 3D chest CT scans from 21,304 patients) and tested on the external RAD-ChestCT dataset (3,630 scans), outperforming CT-CLIP (contrastive language-image pre-training) and CT-Net (full supervision). Code: https://github.com/JasonW375/Chest-OMDL

**Keywords:** Radiology report, Chest computed tomography, Weakly supervised learning, Multidisease detection

## 1. Introduction

Chest computed tomography (CT) is widely used in clinical practice for diagnosing causes of signs or symptoms of chest diseases, such as cough, shortness of breath, chest pain, or fever. CT played a crucial role in the fight against COVID-19 (Ma et al., 2021; Carter et al., 2020). However, each 3D chest CT volume comprises millions of voxels and exhibits significant variations in individual characteristics and imaging conditions, making it cumbersome for radiologists to examine CT volumes slice by slice (Khanna et al., 2020)). To support clinical diagnosis and decision-making while reducing radiologists' workload, many deep learning (DL) models have been developed to assist in CT interpretation and received FDA

---

[*] Contributed equally

[†] Corresponding author

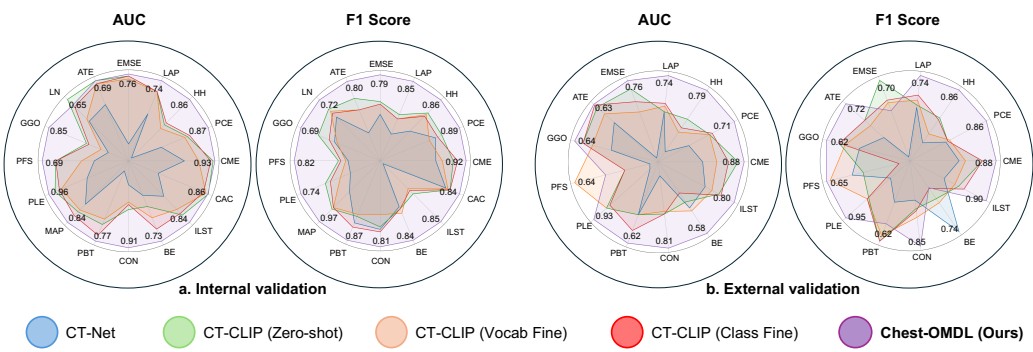

Figure 1: AUROC and F1 scores of Chest-OMDL and comparative methods on internal and external validation datasets. Each dimension of the radar chart represents a specific disease (Abbreviations are defined in Appendix B, detailed results of different methods are provided in Appendix E).

approvals. A research report, Imaging AI 2024 (Miliard, 2024), shows how the number of FDA-approved AI tools for imaging has ballooned to more than 300 in just the past few years, with little sign those approvals will slow.

However, most existing DL models are trained on limited datasets and target only a single disease, raising concerns about the generalizability and robustness of these models in clinical practice (Yang et al., 2024; Chen et al., 2024). Achieving generalizable multidisease detection and lesion localization in chest CT requires large labeled datasets, which are difficult to obtain because annotating medical images is time-consuming and cost-intensive (Rajpurkar et al., 2022; Liu et al., 2024, 2023b; Cao et al., 2022, 2024; Xu et al., 2022; Huang et al., 2023; Fang et al., 2024).

Because ample training data already exist in electronic health records, an alternative approach is to extract information from more accessible free-text radiology reports to train DL models. This primarily involves training contrastive language-image pretraining (CLIP) frameworks (Radford et al., 2021) on large-scale paired image-text datasets for zero-shot classification or applying natural language processing (NLP) techniques (Bergomi et al., 2024) to extract classification labels from text reports for supervised learning. A representative study of the former is CT-CLIP (Hamamci et al., 2024), which was trained with CT-RATE, a large-scale dataset comprising 3D chest CT scans and paired text reports. Hamamci et al. demonstrated that, in multi-abnormality detection, CT-CLIP outperformed state-of-the-art (SOTA) fully supervised models across all key metrics. A notable example of the latter approach is the multidisease classifiers for body CT scans developed by Tushar et al., designed for three different organ systems using automatically extracted labels from radiology text reports (Tushar et al., 2021). Their main contribution involved employing rule-based algorithms to extract 19,225 disease labels from 13,667 body CT scans. Furthermore, Sato et al. (Sato et al., 2024) recently developed a DL-based pipeline to detect abnormalities in the liver, gallbladder, pancreas, spleen, and kidneys, also leveraging information from free-text radiology reports rather than manual annotations. However, the common limitation of the aforementioned methods is that they only perform disease classi-

fication while cannot localize abnormal locations, results in poor interpretability for clinical use. Liu et al. proposed a cross-modality learning framework Cross-DL (Liu et al., 2023a) for detecting four abnormality types across 17 regions in head CT with voxel-level localization. But applying this method to chest CT is challenging due to its large 3D coverage, millions of voxels, and significant variability in imaging conditions.

In this study, to effectively utilize information from free-text radiology reports and achieve simultaneous organ-specific multidisease detection and localization, we propose a novel weakly-supervised learning framework Chest-OMDL for chest CT. Specifically, Chest-OMDL leverages classification labels extracted by RadBERT (Yan et al., 2022) from text reports and segmentation masks generated by the Segment Anything by Text (SAT) model (Zhao et al., 2024) from CT images as weak supervision to train a Y-shaped mamba model (Y-Mamba). The Y-Mamba consists of a feature extractor, an organ segmentation decoder, and a disease anomaly map generator, producing organ segmentation results and lesion heatmaps for chest CT. By incorporating anatomical prior knowledge of each disease during training, the trained Y-Mamba generates interpretable pixel-level lesion localization.

We trained Chest-OMDL on CT-RATE (Hamamci et al., 2024), the largest publicly available chest CT dataset, and compared it with existing methods on both an internal validation set and an external validation set (RAD-ChestCT) (Draelos et al., 2021). Chest-OMDL achieved SOTA performance on multidisease classification tasks across 9 organs. The radar plot in Fig. 1 visually compares the classification AUROC and F1-score of various models. Furthermore, we quantitatively evaluated the localization performance of Chest-OMDL on an external COVID-19 CT dataset (Ma et al., 2020). Despite relying only on organ-level weak supervision, the model achieved a pixel-level segmentation Dice Similarity Coefficient (DSC) of 0.450.

## 2. Materials and Methods

### 2.1. Training and Internal Validation Datasets

We utilized the recently curated and open-sourced CT-RATE dataset by Hamamci et al. (Hamamci et al., 2024) as the training and internal validation dataset, which includes 25,692 non-contrast 3D chest CT volumes along with paired radiology text reports from 21,304 individual patients from Istanbul Medipol University Mega Hospital. The CT-RATE dataset is divided into two groups: 20,000 patients (24,128 volumes) for training and 1,304 (1,564 volumes) for validation. CT volumes are stored in multiple matrix sizes (65.4% at 512×512 pixels, 4.2% at 768×768 pixels, 30.4% at 1024×1024 pixels). The pixel spacing in the axial (XY) plane ranges from 0.227 to 1.416 mm, with a mean of 0.605 mm. The slice thickness varies from 0.035 to 6 mm, with a mean of 1.231 mm. All CT volumes were first resized to 128×128×64 voxels, followed by windowing and histogram equalization for image enhancement.

### 2.2. External Validation Datasets

To evaluate the performance of different methods on out-of-distribution (OoD) data, we utilized the RAD-ChestCT (Draelos et al., 2021) misaligned external validation dataset, which includes 3,630 non-contrast chest CT volumes uniformly reconstructed using a single

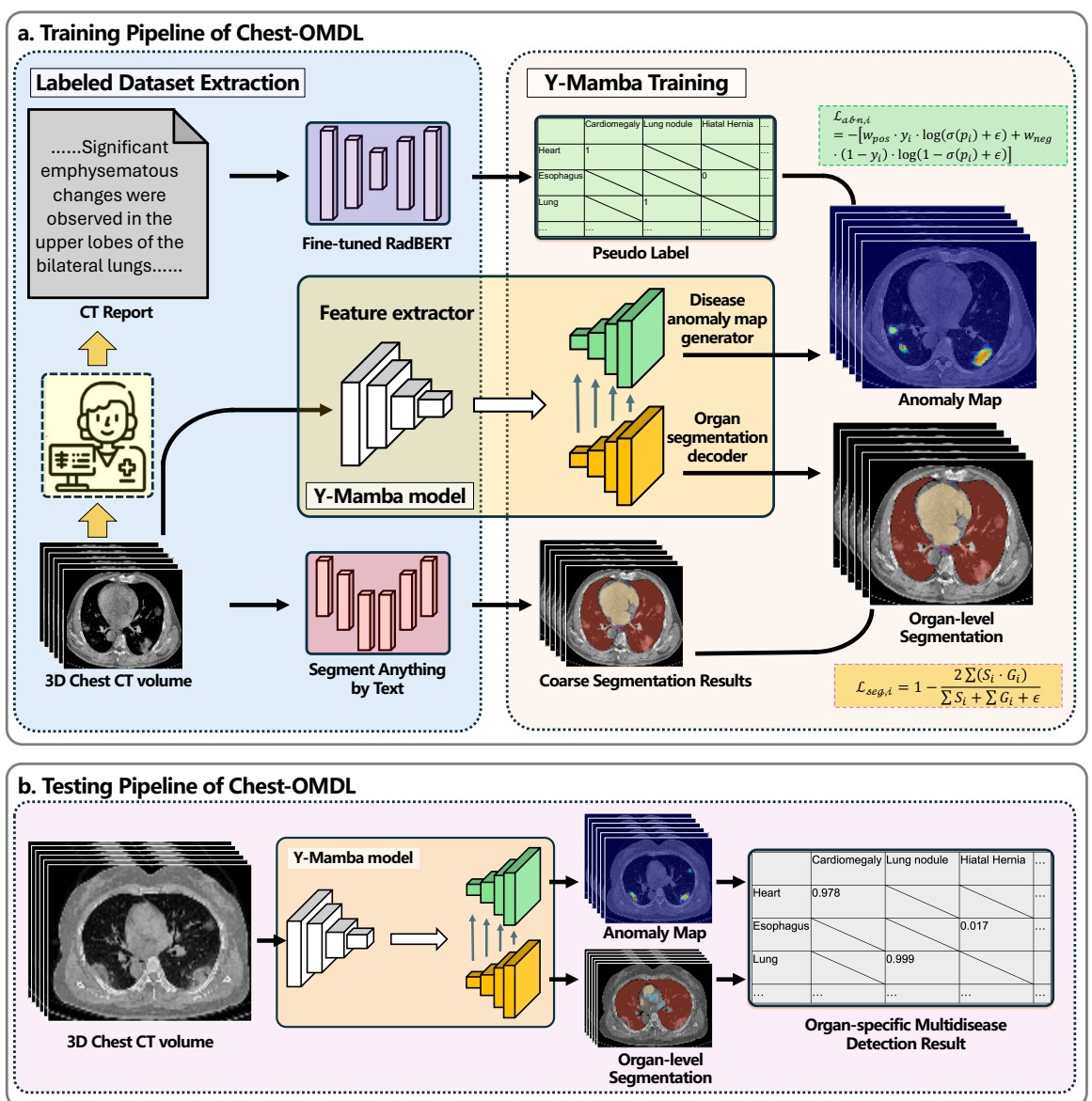

Figure 2: Overview of the proposed Chest-OMDL pipeline.

technique from the Duke University Health System. They have a matrix size of $512 \times 512$ pixels, with axial (XY) pixel spacing ranging from 0.189 to 0.977 mm (mean: 0.692 mm) and slice thickness varying between 0.125 and 5 mm (mean: 0.706 mm). Since neither CT-RATE nor RAD-ChestCT includes lesion segmentation labels, we further evaluated the localization ability of Chest-OMDL using 10 labeled COVID-19 CT scans from the external COVID-19 CT dataset (Ma et al., 2020), where the infections were annotated by two radiologists and verified by an experienced radiologist. To ensure consistent evaluation, we apply identical preprocessing methods to the external datasets as used with CT-RATE.

## 2.3. Chest-OMDL Pipeline

The overall pipeline of the proposed Chest-OMDL is illustrated in Figure 2. The training process consists of two main steps: (1) Labeled Dataset Extraction: Automatically extracting meaningful supervisory information from both text reports and 3D CT images. (2) Y-Mamba Training: Training the Y-Mamba model using a multi-task loss function.

**Labeled Dataset Extraction:** Chest-OMDL assumes that a sufficiently large dataset can make the model robust to noisy labels (Rolnick et al., 2017; Karimi et al., 2020), enabling the direct use of outputs from existing automated methods as training labels. To extract disease labels from CT reports, pre-trained language models can be fine-tuned. In constructing the CT-RATE dataset, Hamamci et al. utilized the RadBERT-RoBERTa-4m model (Yan et al., 2022) to identify disease labels from free-text reports, which we adopt in this study. These labels are linked to six specific organs, creating a tabular pseudo-label for each subject (detailed associations between diseases and organs are in Appendix B). As shown in Supplementary Table 2 of (Hamamci et al., 2024), results from a small-scale test set of 1000 manually annotated reports indicate that RadBERT achieved an average precision of $0.978 \pm 0.024$, a recall of $0.974 \pm 0.027$, and an F1-score of $0.976 \pm 0.016$ across various diseases. This high overall performance makes it well-suited for model training. For segmentation labels, we use the recent Segment Anything by Text (SAT) model (Zhao et al., 2024), a knowledge-enhanced approach leveraging natural language prompts to segment 3D medical volumes. Zhang et al. extended CT-RATE by introducing RadGenome-Chest CT (Zhang et al., 2024), which includes SAT-based segmentation results. We directly use this dataset, retaining segmentation masks for six disease-related organs (Lung, Trachea and Bronchie, Pleura, Mediastinum, Heart, Esophagus) out of the nine available regions.

**Y-Mamba Training:** To achieve multidisease classification and pixel-level lesion segmentation under the weak supervision of organ-level disease localization, we constructed a Y-Mamba model based on the architecture of SegMamba (Xing et al., 2024) (details of the structure are provided in Appendix A). This model simultaneously predicts a segmentation mask for each anatomical organ (from organ segmentation decoder) and an anomaly map for each disease (from disease anomaly map generator). We constructed a multi-task loss function for training Y-Mamba, starting with the Dice loss for the segmentation task:

$$\mathcal{L}_{seg,i} = 1 - \frac{2\sum(S_i \cdot G_i)}{\sum S_i + \sum G_i + \epsilon} \tag{1}$$

where $S_i$ is the predicted segmentation, $G_i$ is the coarse segmentation of the i-th organ generated using SAT, and $\sum(S_i \cdot G_i)$ represents the overlapping pixel count. The totals $\sum S_i$ and $\sum G_i$ correspond to the predicted and ground truth pixel counts, respectively, while $\epsilon$ is a small constant to prevent division by zero. We further define the abnormality detection loss as:

$$\mathcal{L}_{abn,i} = -\left[w_{pos} \cdot y_i \cdot \log(\sigma(p_i) + \epsilon) + w_{neg} \cdot (1 - y_i) \cdot \log(1 - \sigma(p_i) + \epsilon)\right] \tag{2}$$

where $p_i$ represents the model's predicted probability for the i-th disease, obtained by averaging the top-$k$ values after element-wise multiplication of the anomaly map and the segmentation mask of the specific organ. $k$ is an adjustable hyperparameter, set to 24 in this study. An ablation study regarding the selection of the $k$ value can be found in Appendix

G. $y_i$ is the corresponding ground truth (1 for positive and 0 for negative samples). The terms $w_{pos}$ and $w_{neg}$ denote the positive and negative sample weights, respectively, which are derived based on the disease frequency to mitigate biases caused by imbalanced datasets. The sigmoid function $\sigma(p_i)$ ensures that predictions are mapped onto a probability space, and $\epsilon$ serves as a numerical stability constant. The two loss functions are combined using dynamically decreasing weights $\lambda$:

$$\mathcal{L}_{total} = \lambda \sum_i \mathcal{L}_{seg,i} + \sum_i \mathcal{L}_{abn,i} \tag{3}$$

During testing (Fig. 2b), the Y-Mamba model produces segmentation masks for organs and anomaly maps for diseases. By applying the element-wise multiplication of segmentation maps and anomaly maps, as described in Appendix B, the mean of the top-$k$ values is computed as the anomaly score for each disease, consistent with the training phase. Final case-level classifications are based on thresholds, while binarized anomaly maps enable pixel-level segmentation (the process for obtaining model outputs is detailed in Appendix I).

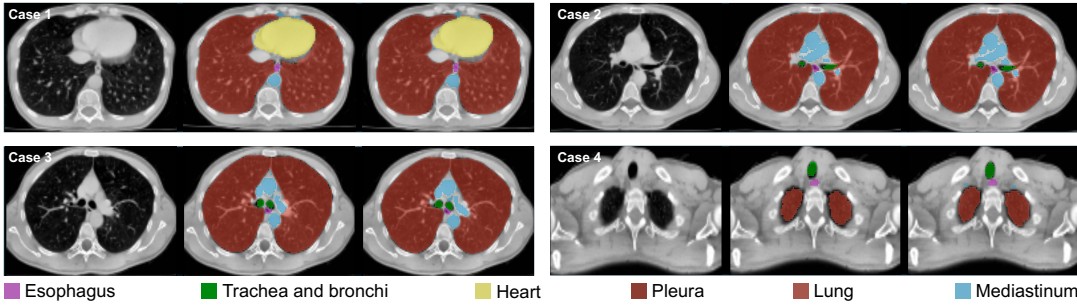

Figure 3: Representative slice-level examples of multi-organ segmentation. From left to right (for each case): the input, the coarse segmentation label generated by the SAT model, and the segmentation mask output by Chest-OMDL.

## 3. Experiments and Results

### 3.1. Training Setup

A total of 22,620 CT volumes from the CT-RATE training dataset were used for training, while the remaining 1,508 volumes were reserved for early stopping and threshold selection. The model with the highest average AUROC on these 1,508 volumes during training was selected, and the threshold was determined using the Jaccard Index. The entire framework was optimized with the AdamW (Loshchilov and Hutter, 2017) optimizer using four NVIDIA A800 GPUs (requiring 2 days 20 hours per training). The number of training epochs was set to 150, with a batch size of 8. The loss function weights were dynamically adjusted during training according to: $\lambda = \max\left(\text{initial\_weight} \cdot e^{-\frac{\text{decay\_rate}}{\text{total\_epochs}} \cdot \text{epoch}}, 0.5\right)$, ensuring gradual decay over the course of the training process. The disease distribution across all datasets is detailed in Appendix C.

### 3.2. Multi-organ Segmentation Results

Accurate segmentation is crucial for Chest-OMDL to identify effective regions in anomaly maps for multiple diseases. The DSC for six organs on the internal validation set are presented in Table 1. Representative slice-level examples are shown in Fig. 3. From left to right (for each case): the input, the coarse segmentation label generated by the SAT model, and the segmentation mask output by Chest-OMDL. For all organs, the average DSC exceeds 84%, with the lungs and pleura achieving the highest DSC values (both at 0.97±0.09), likely due to their highly overlapping anatomical structures. In contrast, the mediastinum has the lowest DSC (0.85±0.09), as its boundaries are indistinct and its intensity falls within the range of normal soft tissue distributions. For comparison, the performance of the specialized organ segmentation model, SegMamba, after training, is detailed in Appendix J.

Table 1: Organ segmentation DSC and NSD metrics on the internal validation dataset (mean ± standard deviation). T&B: Trachea and bronchi.

| Metric | Lung | T&B | Pleura | Mediastinum | Heart | Esophagus |
|---|---|---|---|---|---|---|
| Dice | 0.97±0.09 | 0.90±0.09 | 0.97±0.09 | 0.85±0.09 | 0.91±0.11 | 0.85±0.10 |
| NSD | 0.98±0.09 | 0.98±0.09 | 0.98±0.09 | 0.95±0.09 | 0.93±0.11 | 0.98±0.08 |

Table 2: The performance of multidisease classification is evaluated on both internal and external validation sets using three key metrics: AUROC, accuracy, and F1 score.

| Dataset | Metric | CT-Net | CT-CLIP (Zero-shot) | CT-CLIP (VocabFine) | CT-CLIP (ClassFine) | (Ours) |
|---|---|---|---|---|---|---|
| | AUROC | 0.628 | 0.723 | 0.749 | 0.751 | **0.807** |
| Internal | F1 score | 0.664 | 0.701 | 0.729 | 0.720 | **0.828** |
| | Accuracy | 0.613 | 0.662 | 0.696 | 0.684 | **0.754** |
| | AUROC | 0.549 | 0.624 | 0.651 | 0.643 | **0.720** |
| External | F1 score | 0.594 | 0.641 | 0.665 | 0.655 | **0.723** |
| | Accuracy | 0.543 | 0.591 | 0.617 | 0.610 | **0.659** |

### 3.3. Organ-specific Multidisease Detection Performance of Different Methods

**Comparative Methods:** We compare Chest-OMDL with four SOTA methods that also utilize information extracted from radiology reports to train DL models.: (1) CT-Net (Draelos et al., 2021), a fully supervised traditional classification model trained directly with disease labels; (2) CT-CLIP (zero-shot) (Hamamci et al., 2024), a visual-language foundation model based on CLIP that automatically learns semantic knowledge through contrastive learning; (3) CT-CLIP (VocabFine) and (4) CT-CLIP (ClassFine), two variants of CT-CLIP fine-tuned using different methods specifically for disease classification tasks. These

comparative models were trained and validated on the same dataset, and the experimental results from Hamamcı et al.'s study (Hamamci et al., 2024) are directly used for comparison.

**Validation Results:** The average AUROC, F1 score, and accuracy for detecting 16 diseases (internal validation) and 13 diseases (external validation) are summarized in Table 2. Chest-OMDL shows significant improvements, with higher mean AUROC (+7.74% internal, +10.60% external), mean F1 score (+13.58% internal, +8.72% external), and mean accuracy (+8.33% internal, +6.81% external) compared to CT-CLIP (VocabFine). While its performance on the external dataset is lower than internal dataset, similar to other methods, Chest-OMDL maintains an AUROC above 0.7, comparable to the in-distribution validation performance of comparison methods, demonstrating strong generalization. ROC curves of Chest-OMDL for each disease are in Appendix D, and detailed results for all methods are in Appendix E.

## 4. Lesion Localization Results

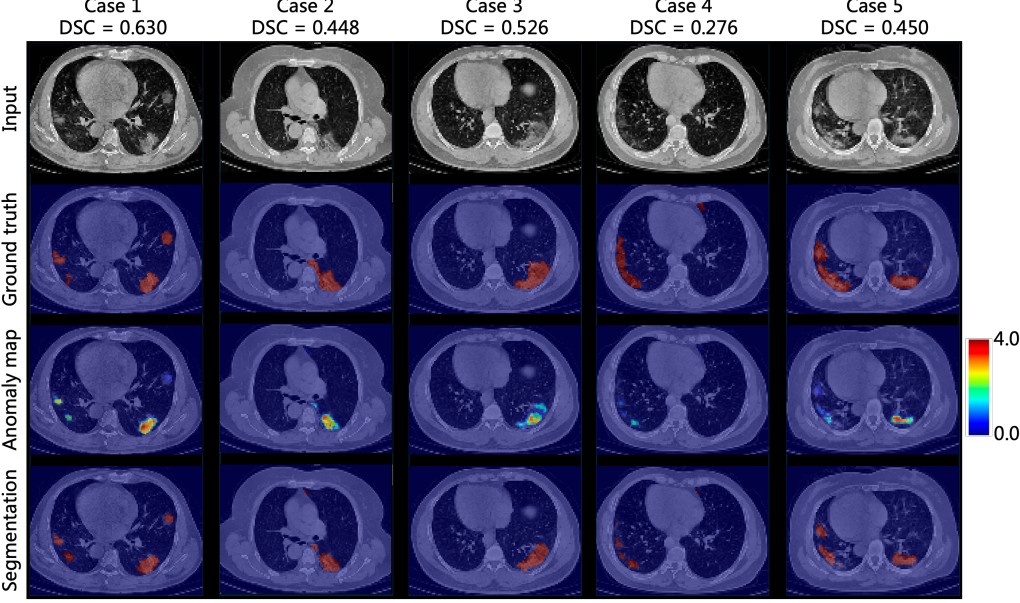

Figure 4: Abnormality localization results on the Covid-19 CT dataset.

We evaluated our model's localization on 10 annotated Covid-19 CT cases (Ma et al., 2020). Covid-19 manifests in CT scans as lung-specific abnormalities like ground-glass opacity (GGO), pulmonary fibrotic sequela (PFS), and consolidation (CON) (Mumoli et al., 2021). To segment these lesions, we overlaid the Chest-OMDL anomaly maps for lung-related diseases (as shown in Appendix B) and used a binarization threshold of 0.1.

Figure 4 shows anomaly maps and segmentation results for five cases, with other cases in Appendix F. Our method achieved an average DSC of 0.450, compared to 0.673 ± 0.223 (Ma et al., 2021) reported by Ma et al. using a supervised approach. This demonstrates that Chest-OMDL, despite being trained with only organ-level weak supervision,

achieved segmentation accuracy equivalent to 67% of that attained by supervised methods. In the transfer learning experiments presented in Appendix H, we further demonstrate that fine-tuning Chest-OMDL with only two subjects can significantly improve segmentation performance.

## 5. Ablation Study

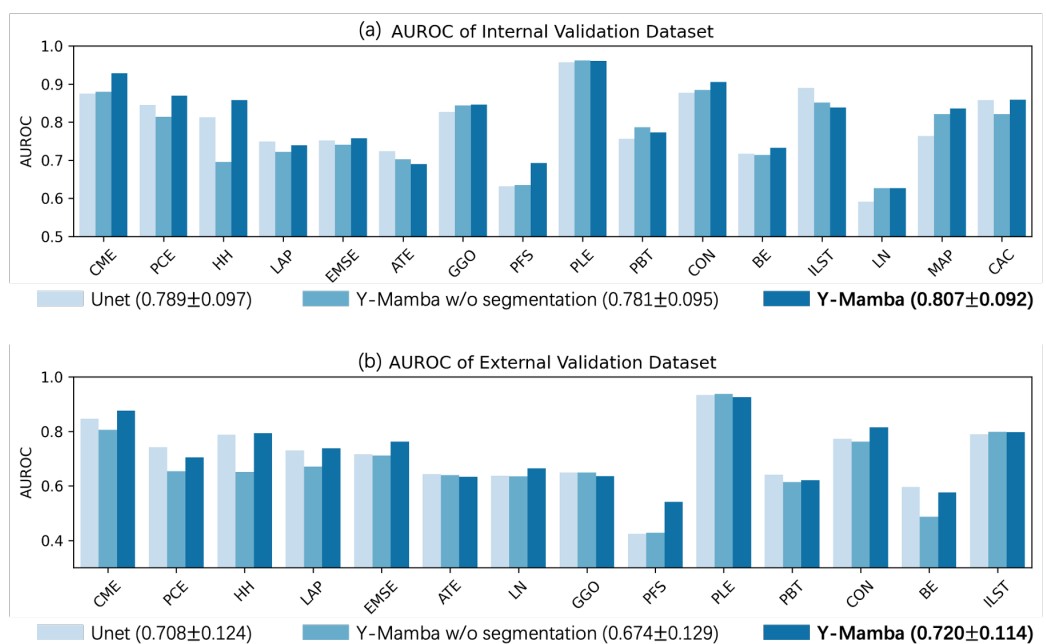

Figure 5: Ablation Study Results. (a) AUROC on the internal validation dataset. (b) AUROC on the external validation dataset.

Chest-OMDL uses weak supervision with labels extracted from a pre-trained model, improving performance through the Y-Mamba architecture and a segmentation task for better classification and localization. Ablation studies compared Unet, Y-Mamba without segmentation, and Y-Mamba (ours) on internal and external validation sets. For Unet (Ronneberger et al., 2015), the final layer was modified to output 6 channels for organ segmentation and 16 for disease anomaly maps. As shown in Figure 5, Y-Mamba achieved the highest average AUROC on both datasets, demonstrating the framework's effectiveness.

## 6. Conclusion

We propose Chest-OMDL, a weakly supervised framework for chest CT disease detection. Using pseudo-labels and anatomical priors, it trains the Y-Mamba model. Chest-OMDL outperforms CT-Net and CT-CLIP on CT-RATE and RAD-ChestCT and demonstrates pixel-level segmentation on a Covid-19 dataset. It reduces costs by eliminating manual annotations for efficient organ-specific diagnosis.

## Acknowledgments

Funding was provided by the Tsinghua University Startup Fund.

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

# Appendix A. The Architecture of the Proposed Y-Mamba

**(a) Y-Mamba model**

**(b) Gated Spatial Convolution (GSC)**

**(c) MambaLayer**

Figure 6: An overview of the proposed Y-Mamba model.

Our proposed Y-Mamba (shown in Fig. 6) is a unified deep learning model designed for organ segmentation and disease anomaly detection in medical imaging. The architecture consists of three main components: a feature extractor, an organ segmentation decoder, and a disease anomaly map generator. The feature extractor encodes multi-scale representations of the input medical images, which are then utilized by the two decoders to perform their respective tasks.

**Feature Extractor:** The feature extraction module adopts a multi-resolution encoding strategy, leveraging 3D convolutional layers (Conv3D) with LeakyReLU activation and Instance Normalization (InstanceNorm3D) to extract spatial and contextual features. To enhance representational power, we incorporate Gated Spatial Convolution (GSC) and Mamba

layers at deeper levels. The downsampling process is implemented via strided convolutions, progressively reducing spatial resolution while increasing feature channels, resulting in hierarchical feature representations.

**1). Gated Spatial Convolution:** The GSC module (shown in Fig. 6(b)) enhances spatial feature learning by combining multiple receptive fields. It consists of parallel 3×3 convolutions, followed by normalization and activation. The outputs are gated through an additional 1×1 convolution that learns spatial importance weights. This mechanism effectively suppresses irrelevant features while enhancing critical spatial patterns.

**2). Mamba Layer:** Each Mamba Layer begins (shown in Fig. 6(c)) with Layer Normalization, which normalizes the input sequence by computing the root mean square value of the input activations. This step is crucial for preventing gradient explosion in deep networks. Following normalization, the Mamba module processes the input sequence, and the resulting output is combined with the input residuals, as expressed in the equation:

$$\mathbf{x}_{i+1} = \text{Unflatten}(\text{Mamba}(\text{Flatten}(\text{LayerNorm}(\mathbf{x}_i)))) + \mathbf{x}_i. \tag{4}$$

Initially, the input features: $\mathbf{x}_i$ undergo a linear transformation and are then split into two components: $\mathbf{y}$ and $\mathbf{z}$. These components are obtained via the operation $\mathbf{y}, \mathbf{z} = \text{split}(\text{linear}(\mathbf{x}_i))$. The $\mathbf{y}$ segment is processed through a 1D convolution, followed by activation and further processing via the Selective Scan Model (SSM):

$$\mathbf{y}_i = \text{SSM}(\text{SiLu}(\text{1D Conv}(\mathbf{y}))). \tag{5}$$

Concurrently, the activated $\mathbf{z}$ segment acts as a gating vector, which is element-wise multiplied with the $\mathbf{y}_i$. Once processed by the Mamba module, $\mathbf{y}_i$ is passed through another linear layer to yield the final result of this module: $\mathbf{x}_{i+1}$.

**Organ Segmentation Decoder:** The organ segmentation decoder follows an encoder-decoder structure, where high-level semantic features are progressively upsampled and concatenated with corresponding low-level features via skip connections. Transposed convolutions (TransConv) are used for upsampling, while multi-layer perceptrons (MLP) further refine the feature representations. The final segmentation mask is generated through a 6-channel output layer, corresponding to the target organs.

**Disease Anomaly Map Generator:** Parallel to the segmentation decoder, the disease anomaly map generator utilizes the same encoded features to predict abnormal regions. Instead of producing organ-wise labels, this branch outputs 16-channel anomaly maps, which highlight potential pathological regions. The decoder follows a structure similar to the segmentation decoder, but with additional feature fusion mechanisms to integrate deeper semantic cues. A final element-wise addition step merges multi-scale outputs to generate the anomaly heatmap.

## Appendix B. Generation of Pseudo Labels

Table 3: The specific associations between diseases and organs.

| | CME | PCE | CAC | HH | LAP | EMSE | ATE | LN | GGO | PFS | PLE | MAP | PBT | CON | BE | ILST |
|---|---|---|---|---|---|---|---|---|---|---|---|---|---|---|---|---|
| Lung | | | | | | ✓ | ✓ | ✓ | ✓ | ✓ | | ✓ | | ✓ | | ✓ |
| Trachea and Bronchie | | | | | | | | | | | | | ✓ | | ✓ | |
| Pleura | | | | | | | | | | | ✓ | | | | | |
| Mediastinum | | | | | ✓ | | | | | | | | | | | |
| Heart | ✓ | ✓ | ✓ | | | | | | | | | | | | | |
| Esophagus | | | | ✓ | | | | | | | | | | | | |

We utilized 16 disease labels extracted from CT-RATE by RadBERT, including Cardiomegaly (CME), Pericardial effusion (PCE), Hiatal Hernia (HH), Lymphadenopathy (LAP), Emphysema (EMSE), Atelectasis (ATE), Lung nodule (LN), Lung opacity (Ground-glass opacity, GGO), Pulmonary fibrotic sequela (PFS), Pleural effusion (PLE), Mosaic attenuation pattern (MAP), Peribronchial thickening (PBT), Consolidation (CON), Bronchiectasis (BE), Interlobular septal thickening (ILST), and Coronary artery wall calcification (CAC). Based on medical domain knowledge, these diseases are associated with different organs to provide coarse localization information. Specifically, as shown in Table 3, lung-related abnormalities are the most frequent, including ATE, LN, GGO, PFS, PBT, CON, BE, and ILST, followed by those related to the heart and trachea. This is because chest CT is primarily used to assess lesions in these regions. During training and inference, anomaly maps are element-wise multiplied by the segmentation mask of their corresponding organ. This constrains the model's attention to relevant anatomical regions, thereby minimizing false positives across organs.

## Appendix C. The Disease Distribution of Different Datasets

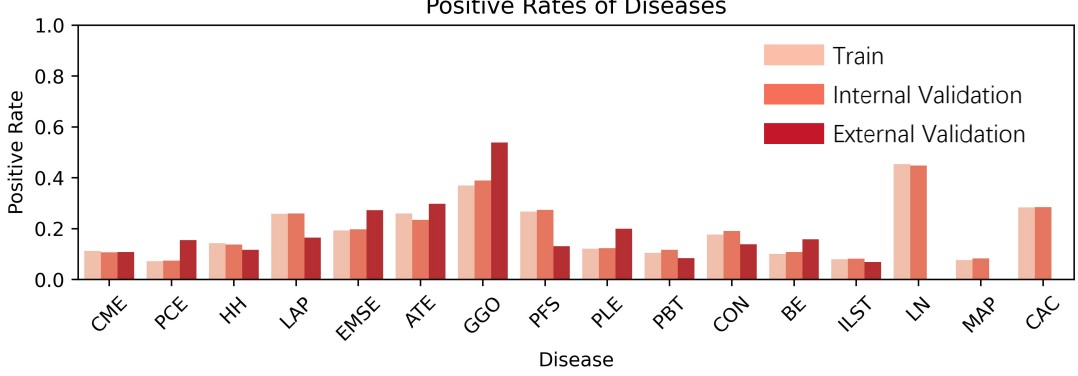

Figure 7: The disease distribution of different datasets (LN, MAP, and CAC are unavailable in the external validation set).

The positive rates of different diseases in the training, internal validation, and external validation datasets are shown in Fig. 7. The data distribution in the training and internal validation sets is consistent, while the positive rates in the external dataset differ significantly from those in the other two sets.

## Appendix D. ROC Curve of Chest-OMDL for Multidisease Detection

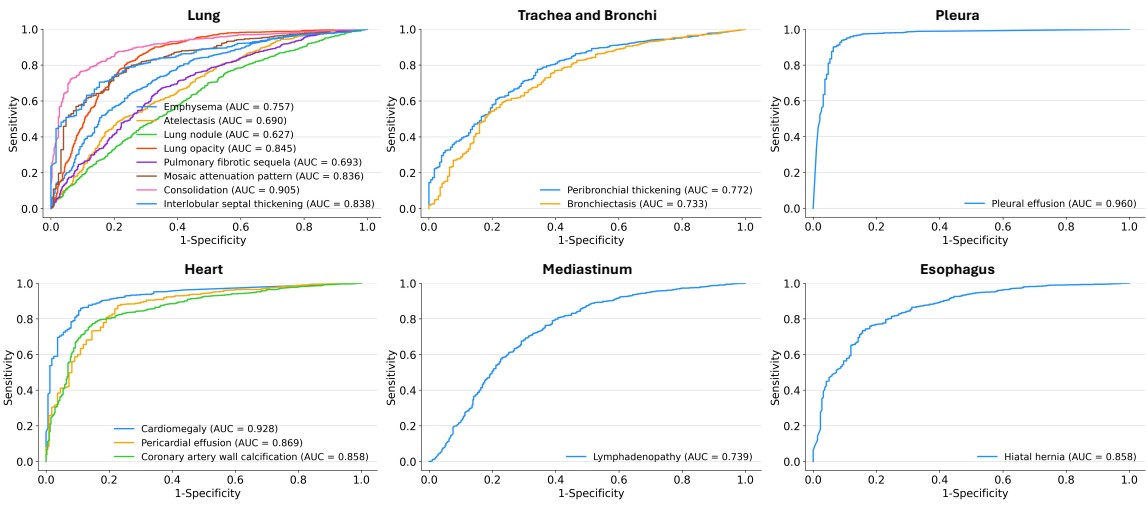

Figure 8: Receiver operating characteristic (ROC) curves for organ-specific multidisease detection (Internal validation dataset).

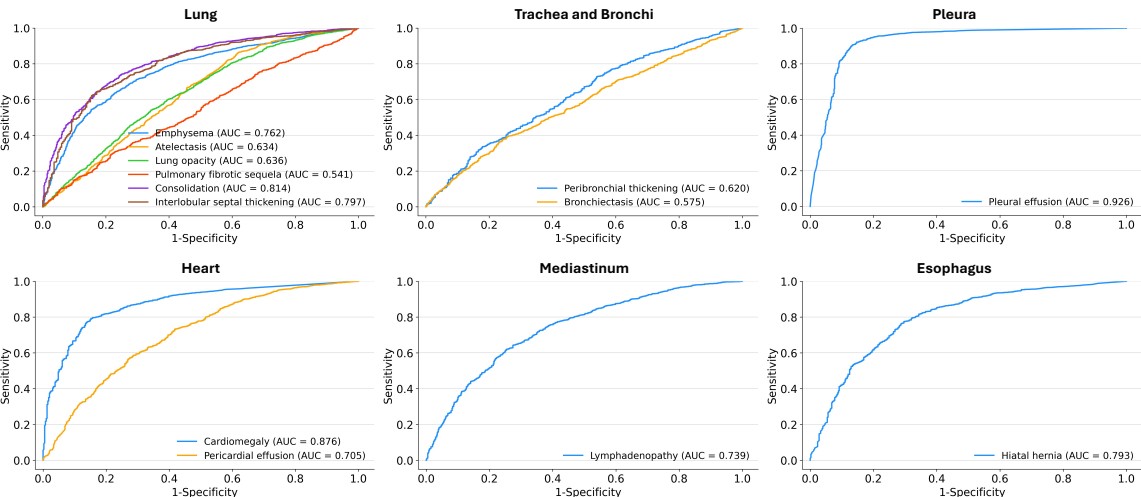

Figure 9: Receiver operating characteristic (ROC) curves for organ-specific multidisease detection (External validation dataset).

# Appendix E. Detailed Detection Performance on Each Disease

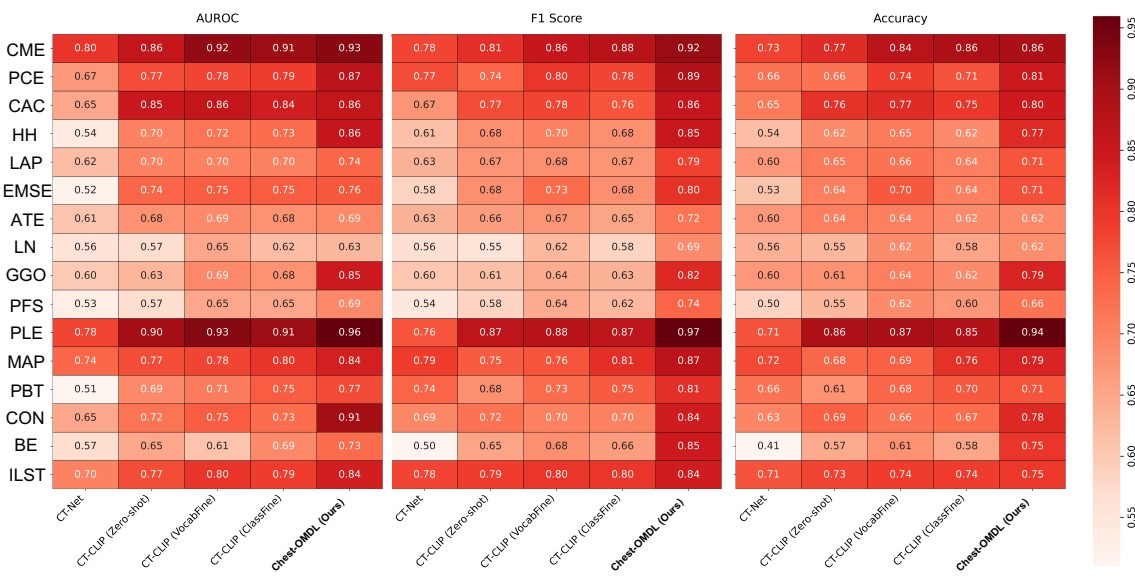

Figure 10: Comparison of anomaly-based performance metrics in the internal validation set.

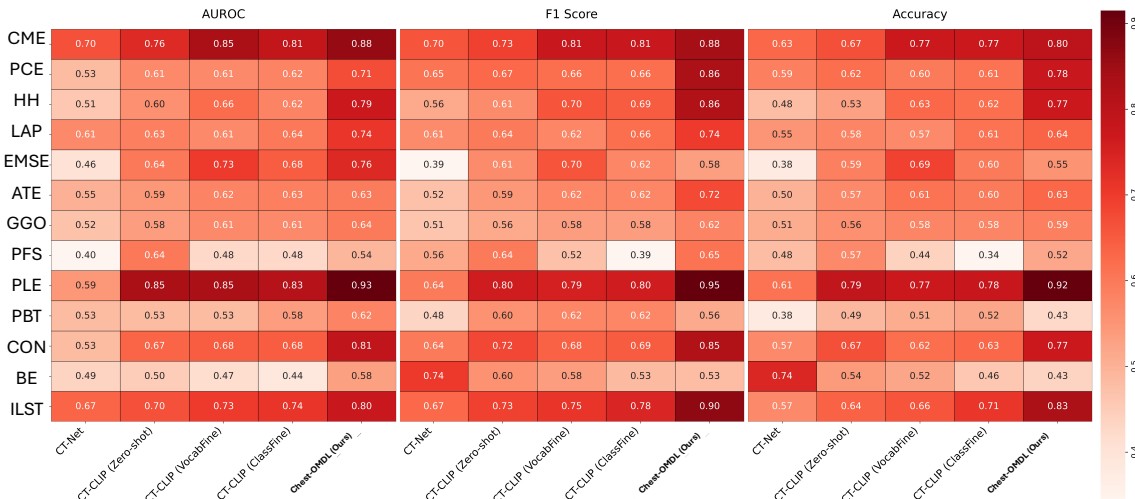

Figure 11: Comparison of anomaly-based performance metrics in the external validation set.

Figures 10 and 11 illustrate the performance of different methods on both internal and external validation datasets across various diseases, evaluated using AUROC, F1 score, and accuracy. The results show that Chest-OMDL outperforms CT-Net, CT-CLIP, and the two fine-tuned variants of CT-CLIP for most diseases. This highlights the model's exceptional adaptability and superior effectiveness under distribution shifts, setting a new benchmark compared to a fully supervised baseline and contrastive learning-based CLIP models.

## Appendix F. Additional Localization Results on the Covid-19 CT Dataset

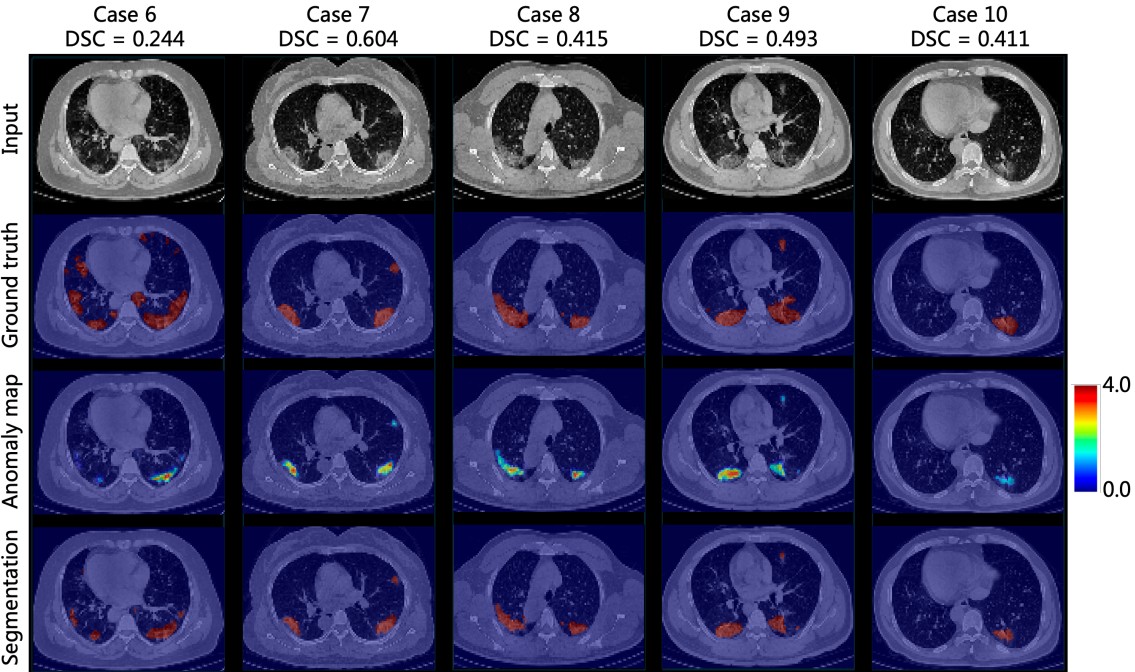

Figure 12: Additional Abnormality localization results on the Covid-19 CT dataset.

Figure 12 shows slice-level examples from other five cases. The suboptimal performance is mainly due to the small size of the abnormal regions localized by Chest-OMDL, which fail to fully cover the lesions, reflecting the model's reduced sensitivity to subtle abnormalities without full supervision. However, across all 10 cases (including Figures 4 and 12), Chest-OMDL consistently detected lung abnormalities and localized the regions of interest, despite imprecise segmentation. Fine-tuning on a small labeled dataset could greatly enhance its performance, which will be a focus of future work.

## Appendix G. Selection of the $k$ hyperparameter

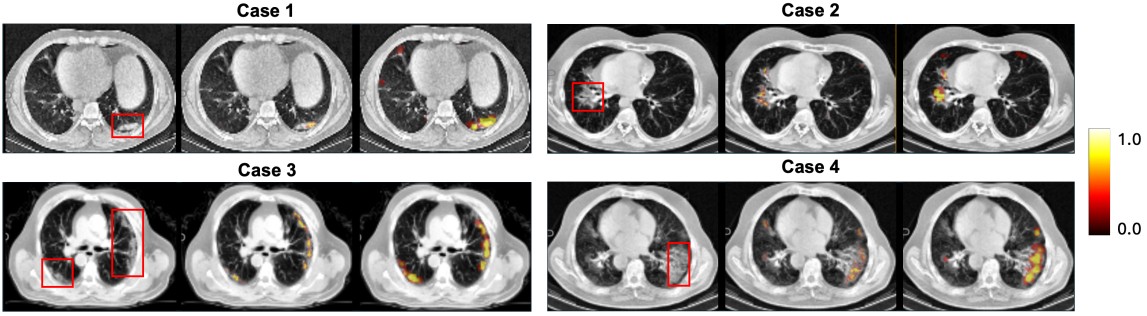

Figure 13: Comparison of anomaly maps generated by single-scale and multi-scale settings (using lung opacity as an example). From left to right: input (with the anomaly location marked by a red box), anomaly heatmap generated by the single-scale setting ($k = 3$), and anomaly heatmap generated by the multi-scale setting ($k = 24$).

The value of $k$ is a hyperparameter that requires optimization on the validation set. We present anomaly heatmaps generated by the model under different settings in Figure 13 (using lung opacity as an example). In the initial phase of model development, we performed experiments using a single scale (i.e., the Disease Anomaly Map Generator outputs only a high-resolution heatmap). We observed that with $k = 3$, anomaly map coverage on the validation set was limited, concentrating primarily on lesion cores (Figure 13). To solve the problem, we implemented a multi-scale approach, leveraging feature maps at varying resolutions during upsampling (integrating a low-resolution scale of $D/2 \times W/2 \times H/2$ with a high-resolution scale of $D \times W \times H$). At the low-resolution scale, k was set to 3 to capture prominent anomalous regions. Recognizing that the highest resolution scale provides an 8-fold ($2^3$) spatial magnification relative to the lowest, we proportionally increased k at the highest scale to $3 \times 8 = 24$ to ensure consistent detection granularity across scales. This design capitalizes on the global contextual information inherent in the low-resolution features while simultaneously expanding the area of anomalous response at high resolution, ultimately improving lesion coverage through multi-scale fusion.

## Appendix H. The Performance of Encoder-only Transfer Learning

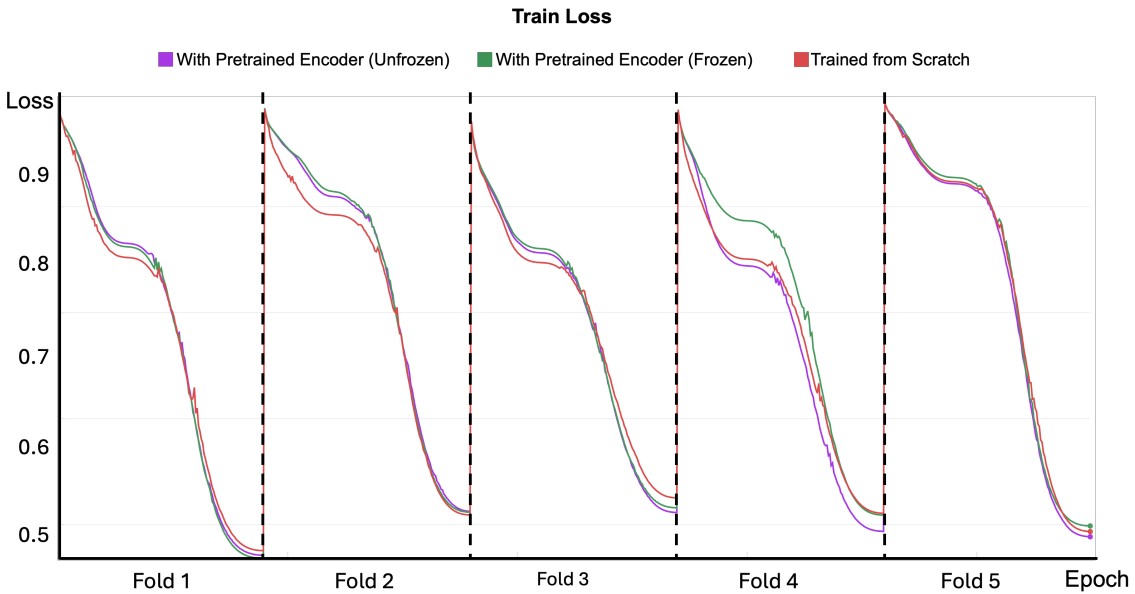

Figure 14: The training loss during the model fine-tuning process.

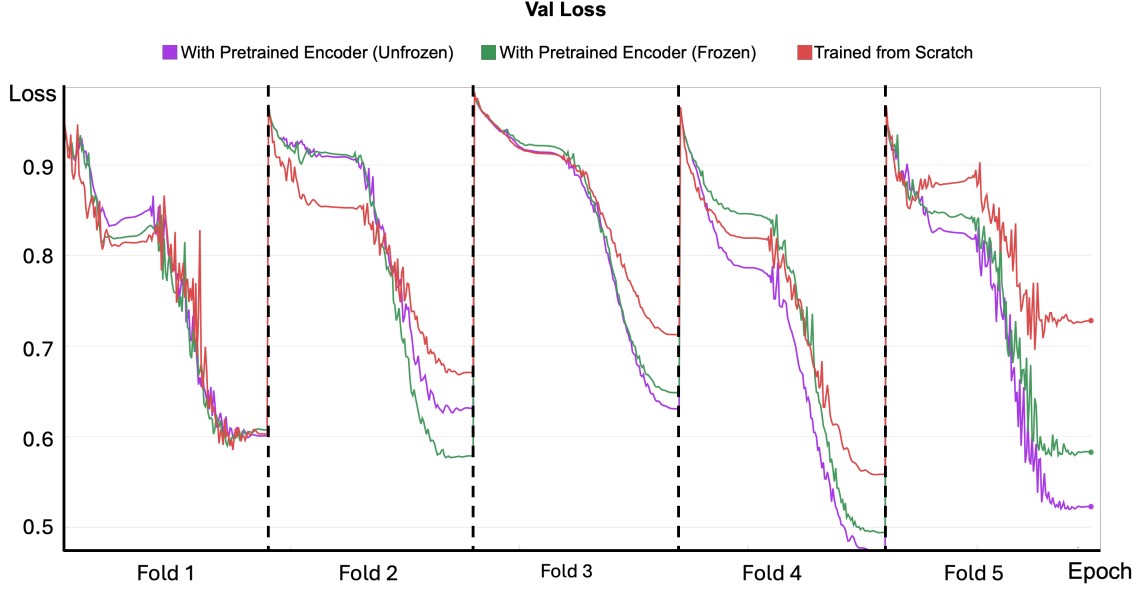

Figure 15: The validation loss during the model fine-tuning process.

To demonstrate the powerful feature extraction capability of the fine-tuned Chest-OMDL encoder, we conducted transfer learning experiments on the Covid-10 CT dataset. Specifically, we constructed a segmentation model, Segmamba, and compared the results of five-fold cross-validation under three settings: (1) The encoder is initialized with Chest-OMDL pre-trained weights and fine-tuned on data from 2 subjects for 150 epochs. (2) The encoder is initialized with Chest-OMDL pre-trained weights and fine-tuned on data from 2 subjects for 150 epochs, with the encoder weights frozen during training. (3) The Segmamba model parameters are randomly initialized and trained from scratch for 150 epochs.

The training loss (on the dataset with 2 subjects) and validation loss (on the dataset with 8 subjects) during the training process are illustrated in Figures 14 and 15, respectively. Although the final training losses of the three settings are similar, the models using pre-trained weights achieved significantly lower validation losses. Furthermore, the segmentation performance on the test set reveals that setting (1) achieved a Dice score of 0.5472±0.0597, setting (2) achieved 0.5512±0.0394, and setting (3) achieved 0.4500±0.0651. It is worth noting that Chest-OMDL's zero-shot segmentation performance already reached a Dice score of 0.450.

These findings support two conclusions: (1) Chest-OMDL requires only few-shot fine-tuning to significantly enhance segmentation performance. (2) The pre-trained encoder demonstrates superior feature extraction and out-of-distribution generalization capabilities for chest CT data.

## Appendix I. Detailed Explanation of Model Outputs

Our proposed Chest-OMDL framework features two output branches:

Disease Anomaly Maps: Each disease corresponds to an anomaly map, where the value of each pixel ranges from 0 to 1, representing the probability of that location belonging to a specific disease's abnormal region.

Organ Segmentation Masks: Binary segmentation of multiple organs, delineating the precise location of each organ.

To achieve organ-specific multi-disease detection, we perform element-wise multiplication of each disease's anomaly map with the segmentation mask of its corresponding organ (e.g., the anomaly map of lung nodule disease is multiplied with the lung organ mask; see Appendix B for the specific disease-organ correspondences). Through this operation, we obtain organ-specific disease prediction maps.

During evaluation, we employ a multiple instance learning strategy, where we select the top-k pixel prediction values from the aforementioned prediction maps and average them, using this average as the final predicted probability of that disease within the entire CT image, thereby completing the disease classification task.

# Appendix J. Organ Segmentation Performance of SegMamba

Table 4: Comparison of Organ Segmentation Performance Between Y-Mamba and Seg-Mamba (mean ± standard deviation). T&B: Trachea and bronchi.

| Model | Lung | T&B | Pleura | Mediastinum | Heart | Esophagus |
|---|---|---|---|---|---|---|
| Y-Mamba | 0.97±0.09 | 0.90±0.09 | 0.97±0.09 | 0.85±0.09 | 0.91±0.11 | 0.85±0.10 |
| SegMamba | 0.97±0.08 | 0.91±0.08 | 0.97±0.08 | 0.87±0.07 | 0.92±0.11 | 0.86±0.10 |

