# OpenReview forum: "Chest-OMDL: Organ-specific Multidisease Detection and Localization in Chest Computed Tomography using Weakly Supervised Deep Learning from Free-text Radiology Report"
_MIDL.io/2025/Conference — MIDL 2025 Poster_

### Official Review · Reviewer_MYaK · 2025-02-19

**Confidence:** 4
**Preliminary Rating:** 3
**Final Rating:** 4

**Summary:**

The paper introduces Chest-OMDL, a framework for multi-disease and multi-organ detection and localization. It utilizes a Y-Mamba model trained with disease labels extracted by RadBERT and organ segmentation masks generated by the SAT model. Experimental results demonstrate its effectiveness in both organ segmentation and disease classification downstream tasks.

**Strengths:**

The paper is well-motivated and well-written. The methodology is clearly explained, the method diagram is well-presented, and the results effectively demonstrate the proposed approach's effectiveness.

**Weaknesses:**

While the paper is well presented and the results show its effectiveness, there are still a few important questions that should be answered.
1.	Since the model learns disease detection and organ segmentation simultaneously, it would be valuable to demonstrate that learning both together outperforms learning each task independently.
2.	The reviewer assumes that the RadBERT and SAT models are not trained during Y-Mamba training. It would be beneficial to show that the proposed method achieves better performance in disease classification than RadBERT and better organ segmentation than SAT.
3.	It would be interesting to evaluate the fine-tuning performance on unseen organs, diseases, and modalities, as well as the zero-shot performance on seen organs and diseases.
4.	The paper would be stronger if the performance of encoder-only transfer learning were reported.

**Detailed Comments:**

Check Weaknesses*.

**Justification Of The Final Rating:**

Thank you for the clarifications and for addressing my main concerns. Since most of my comments have been addressed, I will adjust my rating to a weak accept, as I believe the paper makes a valuable contribution to the field.

**Justification Of The Preliminary Rating:**

The paper is well-motivated, well-structured, and well-written. The technical details are clearly presented, and the results effectively demonstrate the method's performance. The rating is primarily based on the weaknesses discussed earlier. If the authors can address these concerns, I would be happy to reconsider my score.

**Questions To Address In The Rebuttal:**

Check Weaknesses*.

---

> ### Author Response · Authors · 2025-03-07
> **Rebuttal for Reviewer MYaK Part I:**
>
> We appreciate the thoughtful comments and constructive criticism provided by the reviewer. Below, we address each concern and question raised. In addition, based on your feedback, we have made substantial revisions to the manuscript. We believe these changes have strengthened the work and made it more impactful.
>
> ### For Weaknesses 1: Since the model learns disease detection and organ segmentation simultaneously, it would be valuable to demonstrate that learning both together outperforms learning each task independently
> As the objective of this study is to classify multiple diseases across various organs, our contribution lies in introducing segmentation tasks to enhance classification performance while providing interpretability to the model's outputs. Therefore, we conducted ablation studies to examine whether the classification task benefits from the inclusion of segmentation. The corresponding results have been added to Section 5 of the revised manuscript. Specifically, incorporating segmentation improved the average AUROC of Y-Mamba from 0.781 to 0.807 on the internal validation dataset, and from 0.674 to 0.720 on the external validation dataset. Detailed explanations for individual diseases can be found in Figure 5 (newly added in the revised manuscript).
>
> However, since organ segmentation is not the primary objective of Chest-OMDL, it is not necessary for its segmentation performance to surpass that of models trained specifically for segmentation.
>
> ### For Weaknesses 2: The reviewer assumes that the RadBERT and SAT models are not trained during Y-Mamba training. It would be beneficial to show that the proposed method achieves better performance in disease classification than RadBERT and better organ segmentation than SAT.
> This study utilizes pseudo-labels generated by the RadBERT and SAT models to provide weak supervision for organ segmentation and disease classification, respectively. However, it is not meaningful to claim that the proposed method outperforms RadBERT in disease classification or SAT in organ segmentation. This is because RadBERT is specifically designed to extract disease classification results from medical text reports, which routinely contain "explicit disease descriptions." Consequently, RadBERT achieves exceptionally high accuracy, with an average precision of 0.978 ± 0.024, recall of 0.974 ± 0.027, and F1 score of 0.976 ± 0.016 [1]. In contrast, the goal of Chest-OMDL is to output both disease classification and localization results directly from chest CT image data, a task that is significantly more challenging and not directly comparable.
>
> Notably, the baseline methods CT-Net and CT-CLIP, which we compare against, were also trained on the same dataset (CT-RATE), where disease labels were provided by RadBERT. Thus, the fact that Chest-OMDL demonstrates superior classification performance compared to CT-Net and CT-CLIP (as shown in Table 2) is sufficient to validate the effectiveness of the proposed method.
>
> Regarding organ segmentation, as we do not have manually annotated ground truth, the true segmentation accuracy of Chest-OMDL cannot be determined. Furthermore, organ segmentation is not the focus of our method. Instead, segmentation is employed to provide organ-level weak supervision for disease localization, enabling disease localization without manually annotated data. As shown in Figure 4, under the zero-shot setting (and on out-of-distribution testing), Chest-OMDL achieves a Dice score of 0.450, which is 67% of the performance achieved by fully supervised methods on in-distribution testing. This result is highly encouraging. In contrast, the baseline methods CT-CLIP and CT-Net completely lack the capacity for disease localization, leading to poor interpretability of their classification outputs.

---

> > ### Author Response · Authors · 2025-03-07
> > **Rebuttal for Reviewer MYaK Part II:**
> >
> > ### For Weaknesses 3: It would be interesting to evaluate the fine-tuning performance on unseen organs, diseases, and modalities, as well as the zero-shot performance on seen organs and diseases.
> > Thank you for your suggestions. We provide the following explanations regarding the evaluation of fine-tuning performance and zero-shot performance:
> >
> > Fine-tuning Performance: We have detailed the fine-tuning performance of the encoder on the COVID-19 CT Lung and Infection Segmentation Dataset in our response to your Weaknesses 4. These experimental results demonstrate that our model possesses strong transfer learning capabilities and can adapt to new data distributions. In the future, we also plan to further explore and validate the encoder's transfer learning ability on other downstream tasks, such as medical report generation. However, due to time constraints, we were unable to obtain effective results for these tasks before the rebuttal deadline. We will supplement these experiments in subsequent research to comprehensively showcase the model's transfer capabilities.
> >
> > Zero-shot Performance: The localization experiments on the COVID-19 dataset presented in our paper actually serve as a case of zero-shot tasks. The Chest-OMDL model was trained on 16 diseases, none of which include COVID-19-related diseases. To achieve COVID-19 lesion localization, we overlaid abnormality maps of diseases related to COVID-19 (e.g., Lung opacity, Pulmonary fibrotic sequelae) to generate localization results for COVID-19 lesions. These results indicate that, using pseudo-labels extracted from medical report texts as supervision, our model can achieve meaningful Dice scores in a completely zero-shot COVID-19 lesion localization task, demonstrating its zero-shot learning capability.
> >
> > It is worth noting that the primary task of our model is the detection of multiple diseases, while the localization of COVID-19 lesions is more of a validation of the model's zero-shot performance. This localization result further illustrates that the model can focus on the correct anatomical structures and pathological features, rather than relying solely on the overall features of the image for classification. This provides a certain level of interpretability. Such interpretability is particularly important for medical AI systems, as it not only enhances clinicians' trust in automated diagnostic results but also offers more possibilities for the practical application of the model.
> >
> > ### For Weaknesses 4: The paper would be stronger if the performance of encoder-only transfer learning were reported.
> > Thank you very much for your suggestions. In fact, one of our key future research directions is to leverage the Chest-OMDL pre-trained encoder for various downstream tasks. We have reported preliminary transfer learning results on the Covid-19 CT dataset, which we have added to the revised manuscript (Appendix H). Specifically:
> >
> > To demonstrate the powerful feature extraction capability of the fine-tuned Chest-OMDL encoder, we conducted transfer learning experiments on the Covid-10 CT dataset. Specifically, we constructed a segmentation model, Segmamba, and compared the results of five-fold cross-validation under three settings: (1) The encoder is initialized with Chest-OMDL pre-trained weights and fine-tuned on data from 2 subjects for 150 epochs. (2) The encoder is initialized with Chest-OMDL pre-trained weights and fine-tuned on data from 2 subjects for 150 epochs, with the encoder weights frozen during training. (3) The Segmamba model parameters are randomly initialized and trained from scratch for 150 epochs.
> >
> > The training loss (on the dataset with 2 subjects) and validation loss (on the dataset with 8 subjects) during the training process are illustrated in Figures 14 and 15, respectively. Although the final training losses of the three settings are similar, the models using pre-trained weights achieved significantly lower validation losses. Furthermore, the segmentation performance on the test set reveals that setting (1) achieved a Dice score of 0.5472 ± 0.0597, setting (2) achieved 0.5512 ± 0.0394, and setting (3) achieved 0.4500 ± 0.0651. It is worth noting that Chest-OMDL’s zero-shot segmentation performance already reached a Dice score of 0.450.
> >
> > These findings support two conclusions: (1) Chest-OMDL requires only few-shot fine-tuning to significantly enhance segmentation performance. (2) The pre-trained encoder demonstrates superior feature extraction and out-of-distribution generalization capabilities for chest CT data.

---

> > ### Comment · Reviewer_MYaK · 2025-03-13
> >
> > Thank you for the clarifications. Weaknesses 1,3,4 have now been addressed. Regarding W2, the reviewer's concern pertains to real-world applications—specifically, if users aim to perform classification tasks, would RadBERT be the more beneficial choice? Likewise, if SAT is preferable for segmentation-only tasks, what would be the practical application scenario for the proposed method?

---

> > > ### Author Response · Authors · 2025-03-14
> > >
> > > Thank you very much for your careful review of our rebuttal. To further clarify the concerns regarding the weaknesses, we would like to address the following two points:
> > >
> > > Regarding RadBERT:
> > >
> > > (1) In clinical settings, RadBERT is not used for disease diagnosis and classification. In our study, RadBERT was employed to extract labels from retrospective medical text reports provided by clinicians after a diagnosis was made. In practice, doctors visually inspect chest CT scans before delivering their diagnosis and report. Therefore, using RadBERT to extract disease labels after a diagnosis has been established is redundant. In contrast, Chest-OMDL can directly classify diseases from CT images without relying on physicians, thereby serving as a diagnostic aid. The sole purpose of using RadBERT in our research was for model training, as direct contrastive learning between text reports and CT images (i.e., CT-CLIP) has shown limited performance in disease classification, as evidenced by Table 2. Thus, we further utilized RadBERT to convert text reports into explicit disease labels, providing better supervision for the model. In summary, the only function of RadBERT is to provide training labels, which holds no significance in practical application scenario.
> > >
> > > (2) Our research is not the first to use RadBERT or other methods to extract labels from text reports for model training. Variants of CT-CLIP (CT-CLIP(VocabFine) and CT-CLIP(ClassFine)) [1], CT-Net [2], and the recently published fVLM study at ICLR 2025 [3] have employed similar methods to extract labels from text reports for model training. Among these methods, Chest-OMDL demonstrates the best performance, outperforming even the latest ICLR 2025 paper, fVLM, in disease classification on both CT-RATE and Rad-ChestCT (Since this ICLR paper was published after the MIDL submission deadline, we did not compare our work with fVLM in the manuscript). Furthermore, Chest-OMDL is the only method capable of achieving pixel-level disease segmentation. In conclusion, Chest-OMDL represents the state-of-the-art among weakly supervised models of this type.
> > >
> > > Regarding SAT: The role of SAT is to provide organ segmentation labels, which are necessary for offering organ-level supervision for different diseases. However, SAT itself lacks the capability for disease classification and segmentation, thus not conflicting with the application of Chest-OMDL.
> > >
> > > Overall, the practical application of Chest-OMDL lies in providing disease classification and lesion localization results from patients' chest CT images, thereby aiding clinicians in making diagnoses—a capability that neither RadBERT nor SAT offers. RadBERT can only extract disease labels from medical text reports, which is meaningless in practice since the diagnosis has already been concluded by the physician. Additionally, SAT can only perform organ segmentation without diagnosing diseases, making its application context unrelated to that of Chest-OMDL.
> > >
> > > [1] Ibrahim Ethem Hamamci, Sezgin Er, Furkan Almas, et al. Developing generalist foundation models from a multimodal dataset for 3d computed tomography. PREPRINT (Version 1) available at Research Square, October 2024. doi: 10.21203/rs.3.rs-5271327/v1. URL https://doi.org/10.21203/rs.3.rs-5271327/v1.
> > >
> > > [2] Rachel Lea Draelos, David Dov, Maciej A Mazurowski, Joseph Y Lo, Ricardo Henao, Geoffrey D Rubin, and Lawrence Carin. Machine-learning-based multiple abnormality prediction with large-scale chest computed tomography volumes. Medical image analysis, 67:101857, 2021.
> > >
> > > [3] Zhongyi Shui, Jianpeng Zhang, Weiwei Cao, Sinuo Wang, Ruizhe Guo, Le Lu, Lin Yang, Xianghua Ye, Tingbo Liang, Qi Zhang, and Ling Zhang. Large-scale and fine-grained vision-language pre-training for enhanced CT image understanding. In Proceedings of the Thirteenth International Conference on Learning Representations, 2025. Available at: https://openreview.net/forum?id=nYpPAT4L3D.

---

> > > ### Comment · Area_Chair_cUx7 · 2025-03-14
> > >
> > > Hi MYaK, Thank you for commenting about the rebuttal. Could you provide your final rating by the discussion deadline (March 14th Anywhere On Earth)?

---

> > > ### Author Response · Authors · 2025-03-15
> > >
> > > Dear Reviewer,
> > >
> > > We kindly ask if our response has addressed your concerns. We would be sincerely grateful if you could reconsider your evaluation.
> > >
> > > Thank you for your time and effort.

---

> ### Comment · Area_Chair_cUx7 · 2025-03-10
>
> MYaK, Hi. Could you add a comment highlighting which of the 4 weaknesses were sufficiently addressed by the authors and, if they weren’t, why?

---

### Official Review · Reviewer_sduo · 2025-02-20

**Confidence:** 3
**Preliminary Rating:** 2
**Final Rating:** 4

**Summary:**

The authors developed a framework called Chest-OMDL which predicts multi-disease classification and multi-organ segmentation. The framework is trained in a weakly supervise manner, by generating pseudo-labels from radiology reports and segmentation priors generated by Segment Anything by Text. The authors report that their framework surpasses comparative methods and shows capability of pixel-level segmentations.

**Strengths:**

* The performance of the framwork is tested on an internal and external validation set
* Figure 2 is a very nice visualization of the proposed method, which increases readability of the method section.

**Weaknesses:**

* It is unclear what the output of the framework is, description in Section 2.3 is unclear.
* The localization performance is evaluated only on 10 samples with DSC. With a gap of over 20% in DSC, I would not call the proposed method "comparable to supervised methods" (statement in section 4). Thus, the localization is more a first step in proof-of-concept stage and not a matured part of the proposed framework.
* The authors' did not include any baseline results in their evaluation for the performance of the segmentation, eg fully supervised segmentation models like nnUNet. For the classification comparative methods are provided.

**Detailed Comments:**

* The statement of FDA-approved AI-based software seems outdated with number from 2023. Are there more recent numbers available?
* Section 2.2: why are not all 20 available CT scans of the covid dataset used? This would increase the test set, as 10 samples are very little for evaluation.
* For localization in section 4, other metrics aside from DSC could be used to show the detection accuracy. It would be nice to see also cases with lower DSC, it could be mixed between highest and lowest DSC to show the full range of results in the main manuscript.
* For reporting segmentation performance (section 3.2 and section 4), it is good practice to include a boundary-based metric aside from an overlap metric (Dice) to the evaluation.
* It is not clear what the final output of the framework is. At the end of section 2.3 (Fig 2b), it is stated that binarizing anomaly maps produces pixel-level segmentation (localization) - this is clear. But it is unclear what the output of the element-wise multiplication of segmentation and anomaly maps produces, especially since Appendix B is about generation of pseudo labels and not producing final outcomes. In the evaluation it seems that the output are classification labels which is not detection (missing location information). Based on the provided manuscript, this leads to confusion.
* minor comments:
- in section 2.2, there is a sentence without beginning
- caption of Figure 3 does not include what the different columsn represent
- usually figures/tables are placed after text mentioning, close to the text section

**Justification Of The Final Rating:**

The authors addressed my questions and concerns during the rebuttal. The provided clarification and additional information (will) improve the final manuscript. Therefore, I have changed my final rating.

**Justification Of The Preliminary Rating:**

Using existing weak labels, such as extracted features from reports and segmentation masks from SAM-based models, is a nice way to avoid manual annotations and their time-consuming and labor-intensive nature. The framework seems like a promising step in that direction, even when the localization performance is not comparable with current fully supervised SotA methods, it provides a proof-of-concept. However, the current version of the manuscript has some weaknesses in terms of clarity and completeness.

**Questions To Address In The Rebuttal:**

* the outputs of the framework (especially classification/detection output)
* missing segmentation baseline
* lack of second boundary-based metric aside from DSC
* evaluation of localization (number of samples and chosen metrics) and the statement about the comparability of performance with supervised methods

---

> ### Author Response · Authors · 2025-03-07
> **Rebuttal for Reviewer sduo Part I:**
>
> Thank you for your comprehensive review and the time you dedicated to understanding our work. Your insightful comments and constructive suggestions are greatly appreciated, as they have helped us refine and improve the quality of our paper. Based on your feedback, we have made substantial revisions to the manuscript. We believe these changes have strengthened the work and made it more impactful.
>
> ### For Comment 1: The statement of FDA-approved AI-based software seems outdated with number from 2023. Are there more recent numbers available?
> Thank you for pointing out that the data we provided was not the most current. Upon further investigation, we have identified more recent numbers. Specifically, a research report, "Imaging AI 2024" [1], shows how the number of FDA-approved AI tools for imaging has ballooned to more than 300 in just the past few years, with little sign those approvals will slow. We have incorporated the relevant updates into the revised manuscript.
>
> ### For Comment 2: Section 2.2: why are not all 20 available CT scans of the covid dataset used? This would increase the test set, as 10 samples are very little for evaluation.
> We appreciate the reviewers' valuable feedback. The COVID-19 CT Lung and Infection Segmentation Dataset contains 20 cases, with 10 originating from the Coronacases Initiative and the other 10 from Radiopaedia. We opted to use the 10 cases from the Coronacases Initiative primarily due to the following reasons:
>
> 1.  Data Quality Discrepancies: The data from the Coronacases Initiative exhibits higher resolution (512×512×D, where D ranges from 200 to 300) and superior image quality. In contrast, the data from Radiopaedia has a resolution of 630×630×D, where D is typically less than 50, the slice thickness is 6mm, and it suffers from significant noise and lower image quality. Given that the average slice thickness of our training dataset is 1.231 mm, we cannot directly test on such data. However, applying transfer learning on low-quality data in the future is a direction worth exploring. Additionally, reducing image quality during training to achieve data augmentation could be a potential approach to address this issue. However, since exploring how to achieve model generalization on low-quality images is beyond the scope of this paper, we have not pursued it further.
> 2. Lesion segmentation is not the primary objective of Chest-OMDL. Chest-OMDL is primarily designed for lesion classification, where it outperforms similar models such as CT-CLIP and CT-Net. Moreover, lesion localization further enhances the interpretability of classification results, a feature not provided by other methods.
>
> Your observation in "Weaknesses" that "localization is more of a proof-of-concept first step rather than a mature part of the proposed framework" is an accurate understanding. The following further clarifies our localization task:
>
> 1. Localization Method: Our localization results are obtained by element-wise multiplication of the anomaly map for each disease with the segmentation mask of its corresponding organ, resulting in an organ-specific disease prediction map. It is important to note that we did not train or fine-tune on the COVID-19 CT Lung and Infection Segmentation Dataset; instead, we treated it as out-of-distribution data and a zero-shot task.
> 2. Handling COVID-Related Diseases: Since COVID-19 is not included among the 16 diseases we detect, we generated the COVID-19 lesion localization results by superimposing the anomaly maps of COVID-related diseases (e.g., Lung opacity, Pulmonary fibrotic sequela).
> 3.  Significance of Localization Results: Despite the absence of supervised training, our localization results achieved 67% of the performance of fully supervised methods, demonstrating the model's capacity for zero-shot learning. This capability not only demonstrates the effectiveness and transferability of our method but also indicates that the model attends to the correct anatomical structures and pathological features, implying a degree of interpretability rather than relying solely on global features for classification.
>
> It is crucial to emphasize that precise segmentation is not the core task of our framework; the localization of lesions primarily serves to demonstrate the interpretability of Chest-OMDL's classification results. Therefore, we did not provide corresponding baseline results. We consider the COVID-19 dataset experiments as a validation of the model's zero-shot learning capabilities and interpretability, rather than a complete demonstration of the framework's maturity. We believe that this capability can enhance clinicians' trust in automated diagnostic results while providing valuable insights for future research.
>
> In addition, we have further added transfer learning experiments using only the encoder in the revised manuscript (see Appendix H), and we found that fine-tuning with only two subjects can achieve a significant segmentation performance improvement.

---

> > ### Author Response · Authors · 2025-03-07
> > **Rebuttal for Reviewer sduo Part II**
> >
> > ### For Comment 3: For localization in section 4, other metrics aside from DSC could be used to show the detection accuracy. It would be nice to see also cases with lower DSC, it could be mixed between highest and lowest DSC to show the full range of results in the main manuscript.
> > Regarding the evaluation metrics, as we mentioned in our response to comment 2, since precise segmentation is not the core task of our research but is used only to demonstrate the interpretability of the proposed method's classification results, we used only Dice as a metric to simply quantify the segmentation performance. It is important to note that the proposed method was not trained on any labeled COVID-19-related datasets, and the dataset used is out-of-distribution data. Therefore, it is expected that the proposed method would perform worse than fully supervised methods. We demonstrated through transfer learning experiments in Appendix H that fine-tuning the encoder of Chest-OMDL with only two subjects can achieve a significant improvement in the Dice metric. This suggests that the proposed method can be regarded as a foundation model for various downstream tasks (similar to CT-CLIP [2]), which is also the focus of our future research.
> >
> > Furthermore, we have provided visualizations of the segmentation results with lower DSC in Appendix F. Due to space limitations, we did not include this figure in the main text, and we appreciate your understanding.
> >
> > ### For Comment 4: For reporting segmentation performance (section 3.2 and section 4), it is good practice to include a boundary-based metric aside from an overlap metric (Dice) to the evaluation.
> > We have included the boundary-based metric, Normalized Surface Distance (NSD), in the organ segmentation results to further validate the accuracy of organ segmentation (Table 1).
> >
> > ### For Comment 5: It is not clear what the final output of the framework is. At the end of section 2.3 (Fig 2b), it is stated that binarizing anomaly maps produces pixel-level segmentation (localization) - this is clear. But it is unclear what the output of the element-wise multiplication of segmentation and anomaly maps produces, especially since Appendix B is about generation of pseudo labels and not producing final outcomes. In the evaluation it seems that the output are classification labels which is not detection (missing location information). Based on the provided manuscript, this leads to confusion.
> > We appreciate the reviewers' valuable feedback. We would like to further clarify the final output of our framework. Furthermore, to prevent reader misunderstanding, we have added Appendix I, which details the process of obtaining model outputs.
> >
> > Our proposed Chest-OMDL framework features two output branches:
> >
> > Disease Anomaly Maps: Each disease corresponds to an anomaly map, where the value of each pixel ranges from 0 to 1, representing the probability of that location belonging to a specific disease's abnormal region.
> >
> > Organ Segmentation Masks: Binary segmentation of multiple organs, delineating the precise location of each organ.
> >
> > To achieve organ-specific multi-disease detection, we perform element-wise multiplication of each disease's anomaly map with the segmentation mask of its corresponding organ (e.g., the anomaly map of lung nodule disease is multiplied with the lung organ mask; see Appendix B for the specific disease-organ correspondences). Through this operation, we obtain organ-specific disease prediction maps.
> >
> > During evaluation, we employ a multiple instance learning strategy, where we select the top-k pixel prediction values from the aforementioned prediction maps and average them, using this average as the final predicted probability of that disease within the entire CT image, thereby completing the disease classification and detection task.
> >
> > Therefore, the primary task of our framework is to achieve multi-disease classification and detection, with organ segmentation serving as an auxiliary task to enhance the accuracy of disease detection. Furthermore, the generated organ-specific disease prediction maps also possess a degree of lesion localization capability, as validated by our localization experimental results on the COVID-19 dataset. This demonstrates that the model indeed attends to the correct anatomical structures and pathological features, rather than relying solely on the image's overall characteristics for classification. This interpretability is particularly important for medical AI systems, as it can enhance clinicians' trust in automated diagnostic results.

---

> > ### Author Response · Authors · 2025-03-07
> > **Rebuttal for Reviewer sduo Part III:**
> >
> > ### For Comment 6: in section 2.2, there is a sentence without beginning
> > Thank you very much for pointing out our typo. We have corrected it in the revised manuscript.
> >
> > ### For Comment 7: caption of Figure 3 does not include what the different columsn represent
> > Thank you for pointing out the issue. We have added new content to the caption of Figure 3.
> >
> > [1] Mike Miliard. Ai is transforming imaging, with fda approvals continuing apace, 2024.
> > URL https://www.healthcareitnews.com/news/ai-transforming-imaging-fda-approvals-continuing-apace. Accessed: 2025-03-07.
> >
> > [2] Hamamci, Ibrahim Ethem, et al. "Developing Generalist Foundation Models from a Multimodal Dataset for 3D Computed Tomography." arXiv preprint arXiv:2403.17834 (2024).

---

> > > ### Author Response · Authors · 2025-03-08
> > > **Rebuttal for Reviewer sduo Part IV:**
> > >
> > > #### Summary: In summary, the primary goal of this study is to achieve multi-disease classification under a weakly supervised setting. Chest-OMDL significantly outperforms state-of-the-art (SOTA) methods, CT-CLIP and CT-Net, in both in-distribution and out-of-distribution performance. Furthermore, due to the incorporation of organ-level localization information, Chest-OMDL demonstrates a certain degree of localization capability, which was absent in previous weakly supervised methods (CT-CLIP and CT-Net). This represents a major innovation, enhancing the interpretability of classification results. Chest-OMDL’s performance cannot be directly compared to currently fully supervised SOTA methods for two reasons: (1) Chest-OMDL is evaluated under out-of-distribution testing, and (2) it does not use any pixel-level supervision for model training. Therefore, achieving 67% of the in-distribution performance of fully supervised models under these circumstances is highly significant. Finally, we have included transfer learning experiments (Appendix H in the revised manuscript), demonstrating that fine-tuning with data from only two subjects can lead to a substantial segmentation performance improvement.

---

> ### Comment · Area_Chair_cUx7 · 2025-03-10
>
> Hi, sduo. Could you comment on whether your concerns were addressed?  I would like to see at least the following aspects addressed:
>
> Segmentation Baseline: could you confirm if the demanded nnUnet segmentation baseline would be for organ segmentation or lesion localization? If it is for organ segmentation, could you justify why that baseline would be important, given that it is part of an intermediate step of the method, and not the main objective? If it is for lesion localization, could you justify why the Ma et al. (2021) is not enough? In any case, was a good justification provided by the authors to not provide a new segmentation baseline to the rebuttal?
>
> Lesion localization: Was the use of only 10 samples for lesion localization well justified? Was the lack of a boundary-based metric in Section 4 (localization of lesions) well justified? Is the conclusion about the scores for lesion localization adequate now?

---

> > ### Comment · Reviewer_sduo · 2025-03-11
> > **Official Comment**
> >
> > Dear authors,
> >
> > Thank you for the clarifications and addressing my comments.
> >
> > Comment 1, 4, 6, 7: This has been well addressed by the authors.
> >
> > Comment 2: I understand now why only 10 samples out of the 20 available samples have been used and with the additional information shared by the authors, it is well justified.
> >
> > I understand that my comment about the segmentation baseline might not have been clear enough, after the comment of the Area Chair. I was referring to a baseline segmentation for organ segmentation, as there is already a lesion localization baseline by Ma et al. 2021 reported. The authors state that “accurate segmentation is crucial for Chest-OMDL to identify effective regions in anomaly maps for multiple diseases” (Section 3.2). Even though it is an intermediate step to achieve organ-specific multi-disease detection, it is a model output (as now clarified in Appendix I) and influences the main objective. Thus, I was wondering, what the performance gap (or upper limit) to a fully supervised segmentation model for the same task would be, which could be shown by a baseline model.
> >
> > Comment 3: Thank you for the clarification why only DSC was used. Although I do understand the page limit, providing results with lower DSC in the appendix feels a bit like “hidding” them, since Case 4 and 5 could have been exchanged with Case 9 and 10, for example. As the localization is still proof-of-concept and can be improved with future work, it is good to also see what is not working yet, especially since it is part of the framework as clearly stated in the paper title.
> >
> > Comment 5: The additional information about the framework output in the appendix is helpful, but in my opinion, this should be clear from the main text as this is crucial to understand the method. As the framework is called Chest-OMDL, Chest Organ-specific multi-disease detection and localization, I still think the current manuscript structure and usage of terms like detection, localization, segmentation and classification are confusing and take multiple manuscript passes to understand objective/outcome, auxiliary tasks and intermediate steps. In my opinion, this reduces readability.

---

> > > ### Comment · Area_Chair_cUx7 · 2025-03-14
> > >
> > > Hi sduo, Thank you for commenting about the rebuttal. Could you provide your final rating by the discussion deadline (March 14th Anywhere On Earth)?

---

> > ### Author Response · Authors · 2025-03-12
> > **Authors' Response**
> >
> > Thank you to the reviewer for carefully reviewing our rebuttal.
> >
> > In response to your further inquiries, we have attempted to address your concerns:
> >
> > 1. Regarding the baseline for organ segmentation, we would like to clarify the following:
> >
> >    (1) Since the organ segmentation labels in the CT-R1ATE dataset do not have ground truth, we cannot obtain true metrics for organ segmentation on this dataset, including DSC and NSD. The metrics provided in the paper are calculated based on the output of Y-Mamba compared to the organ segmentation results generated by the SAT model [1] which was used in the CT-RATE dataset. Given that the SAT model was trained on 22K 3D medical image scans, we consider it a reasonable approximation of ground truth.
> >
> >    (2) During the model development process, we indeed trained a model specifically for organ segmentation, utilizing the advanced segmentation model SegMamba [2], which is set to be published at MICCAI 2024, as its backbone. The segmentation performance (DSC) for various organs is as follows: Lung - 0.9722 ± 0.0849, Trachea and Bronchi - 0.9082 ± 0.0821, Pleura - 0.9722 ± 0.0849, Mediastinum - 0.8682 ± 0.0742, Heart - 0.9209 ± 0.1087, Esophagus - 0.8636 ± 0.0966. These DSC metrics are indeed slightly better than those of Y-Mamba, though the differences are minimal. To address the reviewer's concerns, we commit to including these results in the final version of the manuscript if the paper is accepted.
> >
> > 2. Regarding Comment 3, we appreciate your suggestion. To alleviate your concerns, we commit to adjusting Figures 4 and 12 in the final version of the manuscript to display both high and low DSC results together. In fact, even the results with low DSC demonstrate highly accurate localization, which supports the excellent zero-shot generalization performance of the proposed method.
> >
> > 3. In response to Comment 5, we acknowledge that the lesion localization results in this study were achieved without using pixel-level labels, thus preventing extremely precise segmentation performance, which we refer to as "localization." However, we have indeed tested its segmentation performance on Covid-19. We recognize that we have mixed the terms detection, localization, segmentation, and classification, which is an area for improvement in the manuscript. If the paper is accepted, we will incorporate the explanation of the Chest-OMDL output into the main text and move some other results (such as parts of the ablation studies) to the appendix to enhance the manuscript's structure. Additionally, we will refine our terminology, using only the terms detection and localization.
> >
> > We hope that our explanations above address the reviewer's concerns.
> >
> > [1] Zhao, Z., Zhang, Y., Wu, C., Zhang, X., Zhang, Y., Wang, Y., & Xie, W. (2023). One Model to Rule them All: Towards Universal Segmentation for Medical Images with Text Prompts. ArXiv, abs/2312.17183.
> >
> > [2] Xing, Z., Ye, T., Yang, Y., Liu, G., & Zhu, L. (2024, October). Segmamba: Long-range sequential modeling mamba for 3d medical image segmentation. In International Conference on Medical Image Computing and Computer-Assisted Intervention (pp. 578-588). Cham: Springer Nature Switzerland.

---

### Official Review · Reviewer_eLre · 2025-02-21

**Confidence:** 5
**Preliminary Rating:** 3
**Final Rating:** 4

**Summary:**

This paper proposes the Chest-OMDL model, a weakly supervised framework that detects and localizes multiple disease in chest CT scans using labels extracted from free-text radiology reports. It employs Y-mamba for organ segmentation and disease detection, reduce the need for manual annotations.

**Strengths:**

Multi-task learning approach: The Y-mamba model integrates both organ segmentation and disease detection, allowing a more comprehensive analysis of chest CT scans.

Chest-OMDL demonstrates better performance than other methods, such as CT-Net.

**Weaknesses:**

1. Dependency on external tools: relies on RadBERT and SAT, which can introduce errors.
2. The explanation of using Y-mamba model is not very clear.
3. The ablation study, such as the mask module, to what extent it contributes to the results is not clear.

**Detailed Comments:**

1. What is the performance of the Rad-BERT on CT disease label extraction?
2. Section. 2.3 says that: p_i represents the model’s predicted probability for the i-th disease, obtained by averaging the top-k values (with k = 24) after element-wise multiplication of the anomaly map and the segmentation mask of the specific organ. Why the k value is set as 24?
3. Section. 2.3. “element-wise multiplication of the anomaly map and the segmentation mask of the specific organ.” Does this mean different disease’s anomaly map will multiply different mask? Because in Table 3. the specific associations between diseases and organs. For example, if having CME anomaly map, the corresponding segmentation mask is only Heart.
4. What is the ablation study result? The result seems good, as you say, segmentation can help classification. It would be better to show to what extent the segmentation result can help the classification task.

**Justification Of The Final Rating:**

The responses provided in the rebuttal, along with the revisions in the manuscript, effectively address the reviewers’ questions and concerns. The authors have conducted sufficient experiments to support their claims, and the manuscript is well-structured and clearly presented. Given the thoroughness of the experimental validation and the clarity of the revisions, this paper should be considered for acceptance.

**Justification Of The Preliminary Rating:**

This paper proposes the Chest-OMDL model, a weakly supervised framework that detects and localizes multiple disease in chest CT scans using labels extracted from free-text radiology reports. It employs Y-mamba for organ segmentation and disease detection, reduce the need for manual annotations. It would be better if the explanation of using the Y-mamba model is provided more clearly and the ablation study is given.

**Questions To Address In The Rebuttal:**

1. What is the performance of the Rad-BERT on CT disease label extraction?
2. Section. 2.3 says that: p_i represents the model’s predicted probability for the i-th disease, obtained by averaging the top-k values (with k = 24) after element-wise multiplication of the anomaly map and the segmentation mask of the specific organ. Why the k value is set as 24?
3. Section. 2.3. “element-wise multiplication of the anomaly map and the segmentation mask of the specific organ.” Does this mean different disease’s anomaly map will multiply different mask? Because in Table 3. the specific associations between diseases and organs. For example, if having CME anomaly map, the corresponding segmentation mask is only Heart.
4. What is the ablation study result? The result seems good, as you say, segmentation can help classification. It would be better to show to what extent the segmentation result can help the classification task.

---

> ### Author Response · Authors · 2025-03-07
> **Rebuttal to Reviewer eLre**
>
> We would like to thank the reviewer for their insightful comments and suggestions regarding our manuscript. We appreciate the time and effort they have dedicated to evaluating our work and are eager to address the concerns raised. In response to the feedback received, we have undertaken substantial revisions.
>
> ### For Question 1: What is the performance of the Rad-BERT on CT disease label extraction?
> Our study employed the publicly available CT-RATE dataset as the training set. The disease labels within this dataset were obtained using a fine-tuned pre-trained RadBERT model. The original CT-RATE paper [1] reported RadBERT's performance on a test set of 1000 manually annotated text reports (refer to Supplementary Table 2 in the paper [1]). Overall, RadBERT achieved an average precision of 0.978 ± 0.024, a recall of 0.974 ± 0.027, and an F1-score of 0.976 ± 0.016 on various diseases. Given its high overall performance, it is suitable for model training. RadBERT's high prediction accuracy can be attributed to the explicit disease descriptions routinely documented in medical text reports. To clarify the sufficiency of RadBERT's accuracy for supporting training, we have incorporated additional details in the revised manuscript (Section 2.3).
>
> ### For Question 2: Section. 2.3 says that: p_i represents the model’s predicted probability for the i-th disease, obtained by averaging the top-k values (with k = 24) after element-wise multiplication of the anomaly map and the segmentation mask of the specific organ. Why the k value is set as 24?
> The value of *k* is a hyperparameter that requires optimization on the validation set. In the initial phase of model development, we performed experiments using a single scale (i.e., the Disease Anomaly Map Generator outputs only a high-resolution heatmap). We observed that with *k* = 3, anomaly map coverage on the validation set was limited, concentrating primarily on lesion cores. To solve the problem, we implemented a multi-scale approach, leveraging feature maps at varying resolutions during upsampling (integrating a low-resolution scale of D/2 × W/2 × H/2 with a high-resolution scale of D × W × H). At the low-resolution scale, *k* was set to 3 to capture prominent anomalous regions. Recognizing that the highest resolution scale provides an 8-fold (2^3) spatial magnification relative to the lowest, we proportionally increased *k* at the highest scale to 3 × 8 = 24 to ensure consistent detection granularity across scales. This design capitalizes on the global contextual information inherent in the low-resolution features while simultaneously expanding the area of anomalous response at high resolution, ultimately improving lesion coverage through multi-scale fusion (as demonstrated by the visual comparison in Figure 13). To clarify this point, we have included additional details in the revised manuscript (Appendix G).
>
> ### For Question 3: Section. 2.3. “element-wise multiplication of the anomaly map and the segmentation mask of the specific organ.” Does this mean different disease’s anomaly map will multiply different mask? Because in Table 3. the specific associations between diseases and organs. For example, if having CME anomaly map, the corresponding segmentation mask is only Heart.
> Yes, your understanding is entirely correct. Our implementation strictly enforces disease-organ anatomical associations as defined in Table 3 (e.g., CME is associated only with the heart; LN, only with lung lobes). Specifically, during training and inference, anomaly maps are element-wise multiplied by the segmentation mask of their corresponding organ. This constrains the model's attention to relevant anatomical regions, thereby minimizing false positives across organs. We have clarified this in the revised manuscript (Appendix B).
>
> ### For Question 4: What is the ablation study result? The result seems good, as you say, segmentation can help classification. It would be better to show to what extent the segmentation result can help the classification task.
>
> The two primary contributions of this study are the introduction of the Y-Mamba model and the enhancement of classification performance through the integration of a segmentation task. To demonstrate effectiveness, we have included the results of ablation studies in the revised manuscript (see Section 5). Specifically, replacing the baseline model Unet with Y-Mamba resulted in an AUROC increase from 0.789 to 0.807 on the internal validation dataset. Furthermore, incorporating the segmentation task into Y-Mamba improved the AUROC from 0.781 to 0.807. Results on the external validation set are detailed in the revised manuscript.
>
> [1] Hamamci, Ibrahim Ethem, et al. "Developing Generalist Foundation Models from a Multimodal Dataset for 3D Computed Tomography." arXiv preprint arXiv:2403.17834 (2024).

---

> > ### Comment · Reviewer_eLre · 2025-03-13
> >
> > I appreciate the authors’ detailed responses and the substantial revisions made to address the raised concerns. The provided explanations clarify key aspects of the study, and the additional details incorporated into the revised manuscript strengthen its rigor.
> >
> > 1. RadBERT Performance on CT Disease Label Extraction: The response effectively justifies the choice of RadBERT and provides strong quantitative evidence from prior research. The clarification regarding the suitability of RadBERT’s performance for model training is helpful.
> > 2. Choice of k = 24 in Probability Computation: The explanation regarding the multi-scale approach and the proportional scaling of k is well-reasoned.
> > 3. Disease-Specific Anomaly Map and Segmentation Mask Multiplication: The inclusion of anatomical constraints to reduce false positives is a strong justification. It might be beneficial to briefly comment on whether this approach introduces any trade-offs (e.g., potential false negatives if an anomaly slightly extends beyond the expected anatomical region).
> > 4. Ablation Study on the Impact of Segmentation on Classification: The numerical results effectively demonstrate the contribution of segmentation to classification performance. The AUROC improvement is notable, and the explanation logically supports the benefits of integrating segmentation. If available, a brief discussion on whether segmentation influences model interpretability (e.g., improved heatmap visualization or lesion localization) could further highlight its value.

---

> > > ### Author Response · Authors · 2025-03-14
> > > **Authors' Response**
> > >
> > > Thank you for your careful reading of our rebuttal. We are pleased to address your concerns. Regarding your additional suggestions, including "briefly comment on whether this approach introduces any trade-offs" and "a brief discussion on whether segmentation influences model interpretability," we indeed observed that removing the segmentation task resulted in diminished performance of the disease localization heatmaps. If the paper is accepted, we will include the relevant content in the appendix.

---

> > > ### Comment · Area_Chair_cUx7 · 2025-03-14
> > >
> > > Hi eLre, Thank you for commenting about the rebuttal. Could you provide your final rating by the discussion deadline (March 14th Anywhere On Earth)?

---

> ### Comment · Area_Chair_cUx7 · 2025-03-10
>
> Hi, eLre. Could you add a comment highlighting if all 4 weaknesses were sufficiently addressed by the authors and, if any individual one wasn't, why?

---

### Author Rebuttal · Authors · 2025-03-07

**Rebuttal:**

# The global rebuttal and the revised manuscript

We sincerely appreciate the reviewers' comments. Based on their feedback, we have made substantial additions and revisions to the paper. Please refer to the revised manuscript in the attached PDF (Supporting Material).

All newly added content has been highlighted in purple text. Specifically, the revised manuscript includes the following updates:

1. In response to Reviewer eLre's Question 1, we supplemented the "Labeled Dataset Extraction" section (Section 2.3) with details on RadBERT's performance in CT disease label extraction.
2. In response to Reviewer eLre's Question 2, we added Appendix G to elaborate on the selection of the hyperparameter *k*.
3. In response to Reviewer eLre's Question 3, we included additional explanations in Appendix B.
4. In response to Reviewer eLre's Question 4, we added ablation study results in Section 5.
5. In response to Reviewer sduo's Comment 1, we updated the description of FDA-approved AI-based software in the introduction.
6. In response to Reviewer sduo's Comment 4, we added the NSD metric for organ segmentation in Section 3.2.
7. In response to Reviewer sduo's Comment 5, we included Appendix I to further clarify the model outputs.
8. In response to Reviewer sduo's Comment 6, we corrected a typo in Section 2.2.
9. In response to Reviewer sduo's Comment 7, we updated the caption of Figure 3 with additional information.
10. In response to Reviewer MYaK's Weaknesses 1, we included ablation study results in Section 5.
11. In response to Reviewer MYaK's Weaknesses 4, we added Appendix H to present transfer learning results.

Additionally, in our initial submission, the font sizes in Table 1 and Table 2 were modified. We have corrected the font sizes in the revised manuscript.

We once again sincerely thank all the reviewers for thoroughly reading our paper and providing constructive feedback, which allowed us to refine the manuscript and strengthen the persuasiveness of the proposed method.

**Supporting Material:**

/attachment/a297cd8930a6bad558cf46aee9de9d2ae16405ea.pdf

---

### Author Response · Authors · 2025-03-13

Dear Area Chairs,

I trust this message finds you well. I am reaching out to kindly seek your assistance regarding the status of the review for my manuscript via OpenReview.

I am grateful for the efforts put forth by the reviewers and the reviewers' community. However, I'm yet to receive feedback on my rebuttal to Reviewer eLre, and Reviewer MYaK, and I believe your guidance could help facilitate the review process.

If possible, could you please consider gently nudging the assigned reviewer or offering any advice that might expedite the review process? Your support would be greatly appreciated, and it would aid in enhancing the overall review timeline.

Thank you so much for considering my request. Your dedication to maintaining the quality of the review process is sincerely valued.

---

### Meta-Review · Area_Chair_cUx7 · 2025-03-20

**Recommendation:** Accept (Poster)
**Confidence:** 5

**Metareview:**

All reviewers recommended weak acceptance. The authors addressed almost all of the concerns of the reviewers with the rebuttal. They are satisfied with the current state of the paper, and highlighted the clear presentation of ideas and robust and thorough experimental section.